# Endothelial Dysfunction in the Pathogenesis of Abdominal Aortic Aneurysm

**DOI:** 10.3390/biom12040509

**Published:** 2022-03-28

**Authors:** Elise DeRoo, Amelia Stranz, Huan Yang, Marvin Hsieh, Caitlyn Se, Ting Zhou

**Affiliations:** Department of Surgery, School of Medicine and Public Health, University of Wisconsin-Madison, Madison, WI 53726, USA; ederoo@wisc.edu (E.D.); astranz@wisc.edu (A.S.); yangh@surgery.wisc.edu (H.Y.); mlhsieh@wisc.edu (M.H.); cse@wisc.edu (C.S.)

**Keywords:** endothelial cell, abdominal aortic aneurysm, endothelial nitric oxide synthase, endothelial cell heterogeneity

## Abstract

Abdominal aortic aneurysm (AAA), defined as a focal dilation of the abdominal aorta beyond 50% of its normal diameter, is a common and potentially life-threatening vascular disease. The molecular and cellular mechanisms underlying AAA pathogenesis remain unclear. Healthy endothelial cells (ECs) play a critical role in maintaining vascular homeostasis by regulating vascular tone and maintaining an anti-inflammatory, anti-thrombotic local environment. Increasing evidence indicates that endothelial dysfunction is an early pathologic event in AAA formation, contributing to both oxidative stress and inflammation in the degenerating arterial wall. Recent studies utilizing single-cell RNA sequencing revealed heterogeneous EC sub-populations, as determined by their transcriptional profiles, in aortic aneurysm tissue. This review summarizes recent findings, including clinical evidence of endothelial dysfunction in AAA, the impact of biomechanical stress on EC in AAA, the role of endothelial nitric oxide synthase (eNOS) uncoupling in AAA, and EC heterogeneity in AAA. These studies help to improve our understanding of AAA pathogenesis and ultimately may lead to the generation of EC-targeted therapeutics to treat or prevent this deadly disease.

## 1. Introduction

Abdominal aortic aneurysm (AAA) is defined as a greater than 1.5-fold increase in the diameter of the aorta beyond the normal range [1]. The dilated abdominal aorta is both structurally compromised and significantly weakened, and consequently often unable to tolerate the forces to which it is exposed from high pressure, pulsatile blood flow. When wall stress exceeds that which the aneurysmal aorta can tolerate, rupture occurs often with catastrophic consequences [2]. AAA rupture is associated with a high overall mortality rate; approximately half of people with ruptured aneurysms will not survive to present to the hospital, and 50% of those who undergo emergency surgery will not survive [3,4]. Risk factors for rupture include large aneurysm diameter and rapid rate of expansion [2]. The annual risk of rupture for patients with aneurysms <5 cm is low, but increases substantially for patients with aneurysms >5 cm (5.0–5.9 cm annual risk of rupture 1–11%, 6.0–6.9 cm annual risk of rupture 10–22%, >7.0 cm annual risk of rupture >30%) [2,5]. Aneurysm size of >5.5 cm in males, >5 cm in females, or rates of growth of >0.5 cm in six months or >1 cm in one year are all indications for repair [5]. Interestingly, AAA exhibits sexual dimorphism, with a 4:1 male to female predominance. While men are more likely to develop AAAs, women have an accelerated rate of aneurysm growth and a higher risk of rupture at smaller aneurysm diameters [6,7,8]. Besides clinical risk factors such as male sex, advanced age, and tobacco use, AAA development is also heavily influenced by genetics [9,10]. Genome-wide association studies (GWAS) have revealed 24 loci associated with increased risk of AAA formation, including genes such as *LRP1*, *LDLR*, *PLTP*, *APOE*, *LPA*, and *PCSK9*, indicating that both family history and individual phenotype influence the onset of AAA [11]. 

The pathophysiology of AAA formation, expansion, and rupture, including its sexual dimorphism, is not fully understood, although existing studies indicate that local inflammation, smooth muscle cell death, and extracellular matrix degradation play a substantial role [2,12]. A feature ubiquitous to nearly all AAAs is a layer of thick intraluminal thrombus (ILT) with associated destruction of the adjacent endothelial lining. While the relationship between ILT and AAA progression has historically garnered interest and been investigated, the consequences of endothelial cell (EC) destruction and endothelial phenotypic changes in the context of AAA have been inadequately investigated [13].

ECs play a major role in regulating vascular homeostasis and blood flow. ECs also modulate vascular tone, angiogenesis, wound healing, smooth muscle cell proliferation, fibrosis, and inflammation. Furthermore, ECs are essential in maintaining a non-thrombogenic blood-tissue interface with limited permeability [14]. Endothelial dysfunction has been shown to initiate various vascular diseases, including AAA [14]. ECs contribute to AAA expansion through increased oxidative stress, nitric oxide (NO) bioavailability, adhesion molecules expression, and inflammatory cell recruitment [15]. In this review, we summarize recent findings on endothelial dysfunction in AAA pathogenesis including clinical evidence of a dysfunctional endothelium in AAA patients, the relationship between biomechanical stress and ECs in AAA, endothelial nitric oxide synthase (eNOS) uncoupling in AAA, and a transcriptional profile of EC dysfunction in AAA revealed by single-cell RNA sequencing (scRNA-seq).

## 2. Endothelial Dysfunction and AAA Progression: Early Clinical Evidence and Common Risk Factors

Currently, AAAs are diagnosed, followed, and intervened upon largely based on abdominal imaging and size-based criteria. The risk of aneurysm rupture increases as aortic diameter increases, from a less than 5% annual risk of rupture for aneurysms 4–4.9 cm in diameter to a 30–50% annual risk of rupture for aneurysms >8 cm in diameter [5,16]. Intervention in the form of endovascular or open surgical aneurysm exclusion is typically offered when AAAs reach a diameter of 5.5 cm in men and 5.0 cm in women, or when growth exceeds 0.5 cm in 6 months or 1 cm in 12 months [5,17]. As our understanding of AAA pathophysiology improves, clinical tests beyond abdominal CT, MRI, and ultrasound and circulating biomarkers may begin to play a role in AAA diagnosis and interventional decision-making. Tests of endothelial function, which have been shown to be abnormal in patients with AAA, are one such promising area [18,19,20,21,22,23].

Flow mediated dilation (FMD) of the brachial artery is a non-invasive measure of endothelial function [22,24,25]. Briefly, FMD indirectly measures the abundance and bioavailability of NO, a soluble gas produced by the endothelium that acts as the main vasodilator within the vasculature [22]. NO is produced in the endothelium by the enzyme eNOS, which catalyzes the conversion of L-arginine into NO and L-citrulline [26]. Circulating NO is able to diffuse through the vascular wall to the medial layer, where it stimulates production of cyclic guanosine monophosphate (cGMP), setting off a cascade of intracellular events that results in smooth muscle relaxation [27,28]. In order to measure FMD in the clinical setting, arterial dilation is induced by inflating and deflating a blood pressure cuff on the extremity of interest, consequently increasing both flow and shear stress and therefore causing an endothelium-dependent release of vasodilating agents (ex: NO) [24]. Ultrasound-based comparisons of the target artery diameter at baseline and after cuff deflation are used to calculate FMD, with a lower FMD indicating endothelial dysfunction [25]. In a cross-sectional observational study of 30 all male patients with CT scan-confirmed AAAs, Medina et al. [18] found a negative correlation between FMD in the brachial artery and aneurysm diameter (R = –0.78, *p* < 0.001). In prospectively recruited cohorts of patients with confirmed or suspected AAA, both Lee et al. [19] and Sung et al. [23] similarly found an inverse correlation between maximum aneurysm diameter and FMD, although of a weaker magnitude than that reported by Medina et al. (Lee et al.: R = –0.28, *p* < 0.001; Sung et al.: R = –0.32, *p* = 0.005). Moreover, Lee et al. found that baseline FMD inversely correlated with aneurysm diameter progression over time and recovered after surgical (n = 22) or endovascular (n = 28) AAA repair [19]. Finally, in a study comparing similarly aged AAA patients (n = 22) and healthy adults (n = 22), Bailey et al. found lower FMD in AAA patients compared to the healthy population (1.10% lower FMD, 95% CI 0.72–0.81) [20].

While promising as a potential sensitive biomarker for AAA diagnosis and progression (Table 1), further investigation is required to better define the specificity of low FMD for AAA. Evidence exists that FMD may be impacted in the setting of peripheral arterial disease (PAD) [29], a pathology showing a moderate association with AAA [30]. Furthermore, Bailey et al. found several baseline differences between AAA patients with lower FMD and healthy peers with higher FMD, namely a higher incidence of hypertension, hyperlipidemia, and poorer peak cardiovascular fitness [20]. Future studies addressing these potential confounders would be of great interest. 

While challenging to study, indirect epidemiological evidence also suggests an intimate link between smoking (the strongest modifiable risk factor for AAA formation), endothelial dysfunction, and AAA development. Tobacco use has been shown to be associated with a 5 to 7-fold increased risk of AAA formation [31,32], and smoking-induced endothelial dysfunction has been well documented over time. Celermajer et al. [33] demonstrated that adult smokers have significantly reduced FMD of the superficial femoral artery and brachial arteries, a finding also reported by Ozaki et al. in a study of young men reported nearly two decades later [34]. The proposed mechanisms by which smoking may drive endothelial dysfunction include generation of reactive oxygen species (ROS) and increased leukocyte adhesion [35,36], with both oxidative stress and vascular inflammation being key pathophysiological events in AAA formation [37,38].

## 3. Endothelial Dysfunction and AAA: Circulating Biomarkers

Beyond investigating FMD as a potential biomarker in AAA, Sung et al. also analyzed circulating endothelial progenitor cell (EPC) burden in AAA patients compared to healthy controls [23]. EPCs are bone marrow-derived cells that circulate in adults and may contribute to endothelial repair [39]. Sung et al. found not only that EPC numbers were reduced in patients with AAA compared to healthy controls, but also that EPC functions (ex: adhesion, proliferation, migration, tube formation) were attenuated in AAA patients [23]. Wu et al. similarly found that pre-intervention number of circulating EPCs were significantly lower in AAA patients compared to healthy controls, and that endovascular aneurysm repair increased EPC numbers as early as two weeks post-repair [40]. While overall a limited number of studies have been performed investigating circulating EPCs in patients with AAA, it is worth noting that one of the earliest studies in this field identified a different pattern with respect to EPCs in AAA patients: Dawson et al. found that AAA patients had significantly more circulating EPCs compared to healthy control patients [41]. These discrepancies may be due to changes in the surface markers used to accurately identify EPCs over time. 

Finally, a limited number of studies have identified several circulating biomarkers for AAA that could be of an endothelial source. Both Ramos-Mozo et al. and Soto et al. found that circulating chemokine (C-C motif) ligand 20 (CCL20), a chemoattractant for lymphocytes and neutrophils that has been implicated in several autoimmune diseases, was elevated in plasma from AAA patients compared to healthy controls [42,43]. Gene expression analysis by Soto et al. revealed significantly higher CCL20 mRNA in human AAA samples compared to control aorta from healthy multi-organ donors. While subsequent immunohistochemical analysis of adjacent AAA tissue revealed that CCL20 localized to both the endothelial and medial layers, the definitive source of circulating CCL20 in AAA patients remains to be determined. Thrombomodulin, an endothelium-bound protein that plays a role in the activated-protein C anticoagulant pathway and has been found to associate with endothelial dysfunction [44], has also been implicated as a potential biomarker in AAA, although conflicting data exists. In a study of 58 male AAA patients and 60 male control patients, plasma thrombomodulin was significantly higher in AAA patients compared to controls [45]. In contrast, a study of 21 AAA patients and 42 healthy controls found no difference in circulating thrombomodulin between groups [46]. 

Until further research is performed, caution should be applied when considering the specificity of these candidate biomarkers for AAA. Studies have shown that CCL20 can be elevated in disease states commonly found in patients with AAA, such as ischemic heart disease and hyperlipidemia [47,48]. Thrombomodulin has also been implicated as a potential biomarker in ischemic heart disease [49] and PAD [44]. Circulating EPC number has also been found to have an inverse correlation with risk factors for PAD [50]. Further research into the sensitivity and specificity of AAA biomarkers of potentially endothelial origin will be merited as our understanding of endothelial dysfunction in AAA pathophysiology continues to grow. 

## 4. Endothelial Dysfunction and Thrombosis in AAA

ILT, while variably sized and shaped, is found within the vast majority of AAAs abutting the intima [51,52,53,54,55]. The ILT observed in AAA is structurally complex and far from biologically inert. Numerous canaliculi of various sizes course from the luminal to abluminal surface, and cellular penetration is observed up to 1 cm in depth from the luminal surface. Erythrocytes, platelets, macrophages, and neutrophils can all be found within the cell-rich regions of the thrombus [53,54]. Conflicting theories have been proposed regarding the role of ILT in AAA pathogenesis. Some posit that ILT accelerates disease progression, while others argue that the thrombus lining decreases wall stress and lowers the risk of rupture. Overall, the current burden of evidence suggests that the prevailing effect of ILT is pathologic, rather than protective, in AAA progression [56]. 

A number of studies have shown that as thrombus volume and thickness increase, the rate of aneurysm growth and risk of rupture both increase [57,58,59]. The sequestered platelets, neutrophils, and macrophages within the body of the thrombus are thought to create a hostile local environment close to the vessel wall that is rich in inflammatory cytokines and proteolytic enzymes [54,56]. Vessel wall hypoxia created by a large luminal thrombus is thought to further contribute to wall weakening and aneurysmal degeneration [55]. ILT has been found to not only correlate with vessel diameter but also with matrix metalloproteinase (MMP) levels, elastin degradation, and smooth muscle cell apoptosis, which are all pathologic hallmarks of aneurysmal degeneration [54,60,61]. Circulating markers of hemostasis have also been found to correlate with AAA and ILT size [62]. Finally, in animal models of AAA, administration of anticoagulants (ex: enoxaparin, fondaparinux) has been shown to reduce intramural thrombus formation and decrease AAA diameter [63]. Of note, studies examining the effect of anticoagulants on AAA progression in humans are lacking. One randomized control trial examining the effect of low dose rivaroxaban plus aspirin in patients with PAD on cardiovascular death, myocardial infarction, or stroke showed reduced major adverse cardiovascular events in the rivaroxaban plus aspirin group compared to aspirin alone [64]. The incidence of AAA in this cohort of patients was not reported. Studies have found that anticoagulation with warfarin after endovascular aneurysm repair for AAA may predispose patients to ongoing type II endoleaks and aneurysm sac growth [65]. A hesitancy to offer anticoagulants to patients at risk for aortic rupture may explain the lack of studies investigating these therapies [66]. 

The arterial endothelium provides a surface for thrombosis formation and regulates blood fluidity and vascular homeostasis. Under conditions of laminar blood flow and high shear stress, healthy endothelium produces a variety of anticoagulant and anti-platelet substances (ex: tissue factor pathway inhibitor, thrombomodulin, NO). However, in the setting of turbulent flow, low shear stress, and endothelial injury (conditions present in the setting of AAA), the endothelial phenotype changes. Under these conditions, ECs shift the balance of their function from anti-thrombotic to pro-coagulant through the binding and activating of platelets and leukocytes, induction of tissue factor, uncoupling of eNOS, and release of von Willebrand factor (VWF), among other actions [67,68,69,70,71]. In a rat xenograft model of AAA, Franck et al. reported that endovascular infusion of ECs prevented AAA formation and stabilized formed AAAs [13]. EC-infusion treated aortas demonstrated reendothelialization, an absence of intraluminal thrombus, the formation of a thick neointima rich in α-smooth muscle actin-positive cells, and a reduction in MMP activity and macrophage infiltration. With respect to mechanism, the authors noted that transplanted rat aortic ECs exerted these effects by paracrine-driven upregulation of endothelial-stabilizing factors and recruitment of native vascular cells, as opposed to directly participating in vessel-wall restoration [13]. Overall, the direct relationship between the endothelium and ILT accumulation within AAAs remains inadequately defined and merits further investigation. 

## 5. Biomechanical Stress on EC

Biomechanical stress plays an important role in the development of various vascular diseases, including atherosclerosis and AAA [72]. Pulsatile blood flow within a vessel exposes the vessel wall to several different biomechanical stresses (axial, longitudinal, shear stress). [73,74]. Shear stress is influenced by blood velocity, viscosity, and vessel diameter [72]. The tangential force that blood exerts along a vessel through flow and friction is referred to as the wall shear stress, a vector measurement influenced by the magnitude and direction of blood flow. [72,75,76,77]. Blood within a vessel maintains a laminar flow pattern and generates high wall shear stress until regions of curvature, branch points, or intraluminal disease are encountered. In these regions, the normally laminar flow pattern is lost and the vessel is exposed to an altered pattern of shear stress [76]. While aneurysms can form anywhere along the aorta, development commonly occurs where normal blood flow is disrupted due to vessel anatomy, such as the curved thoracic aortic arch or infrarenal abdominal aorta immediately proximal to the aortic bifurcation [77,78]. 

In the infrarenal abdominal aorta, the aortic bifurcation generates areas of low oscillatory flow and reduced wall shear stress, along with high shear stress gradients. Continued growth of an existing aneurysm can increase the regions of the wall experiencing low shear stress and local re-circulation of flow [78]. AAAs have been shown to occur when wall shear stress is low but a large wall shear stress gradient exists [79]. In a study conducted by Boyd et al., areas of rupture in AAA had low wall shear stress and thrombus formation, reinforcing that areas with low wall shear stress are prone to aneurysm development [80]. Given how sensitive ECs are to alterations in wall shear stress, it is important to consider how biomechanical stress influences EC function in the context of AAA. 

The EC monolayer serves a crucial role in the regulation and transportation of molecules to other layers of the vessel. Aberrant conditions (disturbed flow or changes in wall shear stress) within the lumen disrupt EC function and can have pathological effects. ECs under disturbed flow conditions undergo a change in morphology, moving from a linear rectangular shape to a circular shape with weakened junctions between cells, increasing the permeability of the EC layer [76,78]. Wall shear stress can be sensed by ECs through receptors facing the vessel lumen, including ion channels, integrins, and G-coupled protein receptors [76,78]. Vascular endothelial growth factor (VEGF) and cell-adhesion molecules VCAM-1 and ICAM-1 have been found to be upregulated in the setting of disturbed flow and low wall shear stress, causing gaps in the EC monolayer and increasing binding of inflammatory cells to both endothelial and vascular smooth muscle cells through gaps in EC junctions [78,81]. Importantly, heightened vascular permeability and inflammation are hallmarks of AAA pathogenesis [82,83]. As a point of interest, several studies have noted sexual dimorphism in human umbilical vein endothelial cells (HUVECs) with respect to baseline expression of genes, including those regulated by shear stress, in addition to differences in the magnitude and direction of transcriptional response to shear stress [84,85,86]. Several studies have also noted sex-based differences in FMD, a measurement that captures vasodilation in response to changes in shear stress [87,88]. Whether or not these different responses to biomechanical stress contribute to differences in aneurysm incidence and rupture risk between men and women remains to be investigated.

## 6. eNOS Uncoupling in AAA

eNOS is a key enzyme involved in the production of NO. The normal function of eNOS requires dimerization of the enzyme and its co-factor tetrahydrobiopterin (BH4). In the absence of BH4, eNOS becomes "uncoupled" and generates superoxide (O_2_–) as opposed to NO, which consequently causes endothelial dysfunction. Bioavailable BH4 levels are determined by a number of factors, including the activity of GTP-cyclohydrolase I (GTPCHI, the rate limiting enzyme in BH4 synthesis), loss of BH4 secondary to oxidation to BH2, and regeneration of BH4 from BH2 by dihydrofolate reductase (DHFR) [89]. Across animal models of AAA, evidence is growing that pathologic eNOS uncoupling contributes meaningfully to aneurysm growth.

The Angiotensin II (Ang II) infusion model is one of the most widely used mouse models of AAA [90]. In this model, Ang II is subcutaneously delivered to apolipoprotein-E null (Apoe^−/−^) mice via osmotic pumps for an extended time, typically 28 days. Certain features of human AAA, including the presence of hyperlipidemia and hypertension, as well as higher incidence in males, are well captured by this model. Other aspects, however, are less well reflected. In the Ang II model, AAAs most commonly develop at the suprarenal rather than infrarenal abdominal aorta, and aortic dissection often precedes dilatation [91]. Using this model, Gao et al. found eNOS and DHFR expression was decreased by Ang II treatment in the intimal layer [92]. The eNOS uncoupling activity, assessed by L-NAME-sensitive superoxide production, was minimal at baseline but greatly exaggerated with Ang II infusion [93]. In cultured bovine aortic ECs, silencing DHFR via RNA interference reduced endothelial BH4 and NO bioavailability [94]. DHFR-deficient mice developed AAAs when infused with Ang II. DHFR-deficient aneurysm-prone mice were also found to have high eNOS uncoupling activity as measured by low BH4 and NO levels, adverse vascular remodeling, and heightened inflammation [95]. This phenotype was rescued by scavenging of mitochondrial ROS with Mito-Tempo [95]. In eNOS pre-uncoupled hyperphenylalaninemia (hph)-1 mice (deficient in GTPCHI), Gao et al. demonstrated that 14 days of Ang II infusion (0.7 mg/kg per day) resulted in 79% AAA incidence and 14% rupture, while none of the wild-type mice infused with Ang II died or developed AAA [96]. BH4 and DHFR were decreased in hph-1 mice, and further reduced by Ang II infusion. Tail vein administration of DHFR expression vector and lipid-based reagent attenuated eNOS uncoupling and prevented AAA formation [96]. Chuaiphichai et al. generated endothelial-specific GTPCH1-deficient mice (Gch1^fl/fl^ Tie2cre). When challenged with Ang II (0.4 mg/kg per day), these mice showed a significant increase in AAA incidence as well as decreased circulating BH4 levels and increased vascular oxidative stress [97]. Administration of folic acid, which was reported to increase DHFR function and recouple eNOS, attenuated AAA development in Ang II-infused Apoe-deficient mice [93]. In contrast, pre-treating C57BL/6 mice with Nω-nitro-L-arginine methyl ester (a NO synthase inhibitor) followed by Ang II and β-aminopropionitrile (BAPN) infusion augmented aortic dissection and aneurysm rupture compared to Ang II and BAPN infusion alone [98]. These studies demonstrated eNOS uncoupling is likely to be involved in AAA progression.

## 7. EC Heterogeneity Revealed by Single-Cell RNA Sequencing

ScRNA-seq has emerged as a powerful and unbiased method in decoding cellular and molecular information in complex tissues by generating transcriptomic profiles of individual cells [99,100]. It has been employed by multiple groups to investigate molecular signatures of human and experimental aortic aneurysm tissue at single-cell resolution (Table 2) [101,102,103,104,105]. As a monolayer covering the aorta, ECs were shown to be a smaller cell population compared to other vascular cell types such as smooth muscle cells and fibroblasts. Of note, the relatively low cell number makes EC analysis challenging, given cell populations with higher numbers will generate the most reliable results with numerous observations to analyze. 

Yang et al. conducted scRNA-seq on aortic tissue harvested from mice exposed to the CaCl_2_ AAA model [104]. Briefly, the infrarenal abdominal aortas of C57BL/6J mice were peri-vascularly treated with 0.5 M CaCl_2_ (AAA group) or NaCl (control group). Aortas were collected four days after AAA induction to capture acute transcriptional responses within the aortic wall. This dataset contained 3896 cells in total, including 2537 cells from the control group and 1359 cells from the AAA group. There were 77 ECs in the control group (3%) and 60 ECs (4.1%) in AAA. Unbiased clustering further identified two EC sub-populations based on the transcriptional features EC1 and EC2. EC1 was enriched in lipid metabolism genes such as *Gpihbp1*, *Fabp4*, *Cd36*, while EC2 highly expressed matrix remodeling genes such as *Mgp*, *Bgn*, *Sulf1*. Among total ECs, there were 61% EC1 and 39% EC2 in the control group, and 78% EC1 and 22% EC2 in the AAA group. Zhao et al. analyzed aneurysmal tissue from the peri-adventitial Elastase model [105]. In this model, infrarenal abdominal aortas from C57BL/6J mice were treated with 30 μL elastase or heat-inactivated elastase (control). Aortas were collected 7 or 14 days after elastase exposure or 14 days after heat-inactivated elastase exposure (control group). Consistent with the CaCl_2_ dataset, two EC sub-populations were also identified in this dataset, with similarly enriched genes. Additional analysis of scRNA-seq data from Ang II infusion-treated Apoe-deficient mice and human AAA samples confirmed the presence of these two EC functional states, suggesting the universal existence of these two EC sub-populations in both human AAA and mouse models of AAA. As spatial information is lost in scRNA-seq analysis, further spatial orientation-preserving validation in mouse and human AAA samples would be necessary and informative, along with a deeper analysis of how these two EC sub-populations contribute to AAA progression. 

## 8. Conclusions

Aneurysmal degeneration of the abdominal aorta is a relatively common phenomenon that can be life threatening when an aneurysm becomes sufficiently large or grows at a rapid rate. Unfortunately, no current therapies exist outside of open surgery or endovascular intervention to halt the progression of aortic degeneration and reduce the risk of rupture. Better understanding of the cellular and molecular pathologic processes that drive AAA formation is the first step in being able to develop non-invasive therapies to treat, or even prevent, aneurysm formation. While the roles of local inflammation, matrix degradation, and smooth muscle cell death have been investigated in the context of AAA, limited attention has been given to the role that endothelial dysfunction may play in AAA development. This review provides an overview of studies showing evidence of endothelial dysfunction as a pathologic contributor to AAA formation across both animal models and in humans. Collectively, these studies show that the mechanisms by which dysfunctional ECs drive AAA formation are diverse, ranging from decreased NO production/eNOS uncoupling to a phenotypic switch that results in a pro-thrombotic, pro-inflammatory state with differing distributions of EC sub-populations in the setting of pathologic biomechanical stresses in aneurysm prone regions (Figure 1). 

This review is not without limitations. The role of EC dysfunction as a contributor to AAA formation has been relatively under investigated, and as such a limited amount of data is available. As further studies focusing on EC dysfunction in the context of AAA become available, a systematic review, as opposed to narrative review, would benefit the field. Moreover, given that endothelial dysfunction is observed in cardiovascular disease beyond AAA, continuing to pursue experimental studies that directly manipulate endothelial genes/phenotypes in the context of AAA models is necessary. This will not only provide further insight into the mechanisms by which ECs contribute to AAA but will also bolster confidence that EC dysfunction is an active contributor to aneurysm pathophysiology. As our understanding of EC contributions to aneurysm formation evolves, our ability to explore endothelial-targeted or protective therapies in the context of aneurysm treatment will also be enhanced.

## Figures and Tables

**Figure 1 biomolecules-12-00509-f001:**
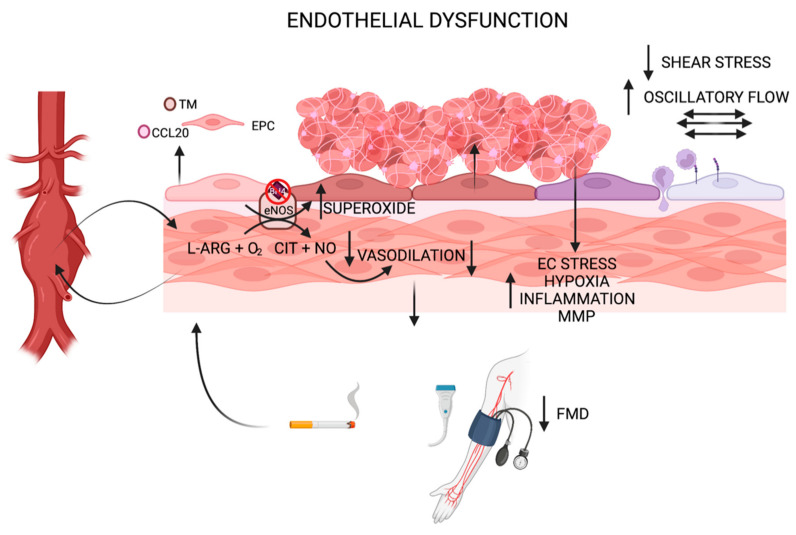
Diagram depicting mechanisms by which endothelial dysfunction contributes to AAA growth, including eNOS uncoupling, a phenotypic switch to a pro-thrombotic, pro-adhesive, permeable state in the setting of altered biomechanical stress, and EC stress generated by intraluminal thrombus accumulation. EC populations 1 and 2 as determined by single-cell RNA sequencing are shown in distinct colors (pink, purple) in low (light) and high (dark) stress states. Potential biomarkers (circulating TM, CCL2, EPCs, and FMD analysis) are depicted. Tobacco use is highlighted as a clinical risk factor for endothelial dysfunction and AAA progression. Abbreviations: AAA (abdominal aortic aneurysm); ARG (arginine); CCL20 (C-C Motif Chemokine Ligand 20); CIT (citrulline); EC (endothelial cell); eNOS (endothelial nitric oxide synthase); EPC (endothelial progenitor cell); FMD (flow mediated dilation); MMP (matrix metalloproteinase); NO (nitric oxide); TM (thrombomodulin).

**Table 1 biomolecules-12-00509-t001:** Clinical studies of flow mediated dilation (FMD) in abdominal aortic aneurysm (AAA).

Authors	Study Design	Participants	Methods	Findings
Medina et al., 2010 [18]	Cross Sectional	N = 30 (30 men)	Correlate FMD with AAA diameter at a static point in time	Negative correlation between AAA diameter and FMD (R = −0.78 *p* < 0.001) FMD significantly differed across AAA diameter quartiles (*p* < 0.001)
Sung et al., 2013 [23]	Cross Sectional	N = 78 (15 healthy controls [100% men], 27 small AAA [93% men], 36 large AAA [89% men])	Evaluate FMD in patients with normal aortic diameter (M < 3.5 cm, W < 3 cm), small aneurysm (M 3.5–5.5 cm, W 3–5 cm), large aneurysm (M > 5.5 cm, W > 5 cm)	FMD was significantly lower in large (5.26 ± 3.11%) and small (6.31 ± 3.66%) AAA patients compared to controls (8.88 ± 4.83%, *p* = 0.008)
Lee et al., 2017 [19]	Prospective Cohort	N = 162 (147 men, 15 women)	Measure AAA diameter and FMD over time	Negative correlation between AAA diameter and FMD (R = −0.28, *p* < 0.001) FMD inversely correlated with AAA diameter progression (R = −0.35, *p* = 0.001) FMD deteriorates over AAA surveillance (median 2% at baseline to 1.2% at follow up; *p* = 0.004) Surgical repair of AAA leads to improved FMD (1.1% pre-op to 3.8% post op, *p* < 0.001)
Bailey et al., 2018 [20]	Prospective Cohort	N = 44 (22 AAA patients, 22 healthy adults, 100% men)	Measure FMD in AAA patients and healthy controls at baseline and after exercise	Baseline brachial FMD was 1.10% lower (95% CI 0.72–0.81) in AAA patients compared to healthy controls

Flow mediated dilation (FMD); Abdominal aortic aneurysm (AAA); M (men); W (women).

**Table 2 biomolecules-12-00509-t002:** Single-cell RNA sequencing studies of abdominal aortic aneurysm (AAA).

Authors	Human AAA or Mouse AAA Model	Control Group	Duration of AAA Induction
Davis et al., 2021 [103]	Tissue from AAA patients undergoing open aortic aneurysm repair	Tissue from patients undergoing open aortobifemoral bypass	Not applicable
Hadi et al., 2018 [101]	1000 ng/kg/min Ang II infusion in Apoe^−/−^ mice	PBS-infused Apoe^−/−^ mice	28 days
Boytard et al., 2020 [102]	1000 ng/kg/min Ang II infusion in Apoe^−/−^ mice	PBS-infused Apoe^−/−^ mice	28 days
Li et al., 2021 [106]	1000 ng/kg/min Ang II infusion in Apoe^−/−^ mice	Saline-infused Apoe^−/−^ mice	28 days
Zhao et al., 2021 [105]	Peri-adventitial incubation of elastase in C57BL/6J mice	Peri-adventitial incubation of heat-inactivated elastase in C57BL/6J mice	7 and 14 days
Yang et al., 2021 [104]	Peri-adventitial incubation of 0.5 M CaCl_2_ in C57BL/6J mice	Peri-adventitial incubation of 0.5 M NaCl in C57BL/6J mice	4 days

AAA (abdominal aortic aneurysm); Ang II (Angiotensin II).

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
