# Peer review of "Endothelial Dysfunction in the Pathogenesis of Abdominal Aortic Aneurysm"

_biomolecules, 2022, doi:10.3390/biom12040509_

Round 1

Reviewer 1 Report

My comments/suggestions are in attachment.

Reviewer 2 Report

I appreciate the efforts of the authors. The topic could be interesting. However a PRISMA should be addded; a systematic review should be preferred. In table 1,  abbreviations and references are lacking. Pleas add. Moreover, the "take home messages" should be more precise. Finally 3-4 figures should be added.

Reviewer 3 Report

This is a nice and well written review

Author Response

The authors thank the reviewer's comment.

Round 2

Reviewer 2 Report

I appreciate the authors' efforts; however, unfortunately, the overall flow is poor and not well organized despite the attempt for corrections. Furthermore, the grammar and syntax are poor.

Author Response

The authors appreciate the suggestions for improvement provided by the reviewer. With respect to the reviewer’s original comments, while we agree a systematic review on this topic would be of great interest as research in this field grows and targeted clinical questions begin to arise, this is unfortunately beyond the scope of our current article. As suggested, the tables were updated to clarify references where listed, and the conclusion paragraph was edited to clarify the take home messages of the review.

With respect to the reviewer’s comments after the first round of revisions, we appreciate the suggestion to refine the flow and syntax of the article, and have edited the article throughout to improve clarity, flow, and syntax (these newly made edits are shown in track changes).